# Association between anxiety and depression and all-cause mortality: a 50-year follow-up of the Population Study of Women in Gothenburg, Sweden

Amanda von Below,[1] Tore Hällström,[2] Valter Sundh,[1] Cecilia Björkelund ,[1] Dominique Hange [1]

[1]Primary Health Care/School of Public Health and Community Medicine, University of Gothenburg Sahlgrenska Academy, Goteborg, Sweden
[2]Department of Psychiatry and Neurochemistry, Gothenburg University, Institute of Neuroscience and Physiology, Sahlgrenska Academy, Gothenburg, Sweden

**Correspondence to**
Dr Dominique Hange;
dominique.hange@allmed.gu.se

## ABSTRACT

**Objectives** This study aimed to examine the association between anxiety disorders and/or major depression disorder (ADs/MDD) and all-cause mortality in a 50-year perspective and to examine specific risk and health factors that may influence such an association.

**Design** Observational population study, 1968–2019.

**Setting** The Population Study of Women in Gothenburg, Sweden (PSWG).

**Participants** In 1968–1969, 899 (out of 1462) women from PSWG were selected according to date of birth for a psychiatric investigation, including diagnostic evaluation. Eight hundred (89%) were accepted. Twenty-two women were excluded. Of the 778 included, 135 participants (17.4 %) had solely ADs, 32 (4.1%) had solely MDD and 25 (3.2%) had comorbid AD/MDD.

**Primary and secondary outcome measures** Associations between ADs, MDD, comorbid AD/MDD and all-cause mortality with adjustments for potential confounding factors. Differences between the groups concerning health and risk factors and their association with mortality.

**Results** In a fully adjusted model, ADs were non-significantly associated with all-cause mortality (HR 1.17, 95% CI 0.98 to 1.41). When examining age during risk time as separate intervals, a significant association between mortality and AD was seen in the group of participants who died at the age of 65–80 years (HR 1.70, 95% CI 1.26 to 2.29). In the younger or older age interval, the association did not reach significance at the 95% level of confidence. Among confounding factors, smoking and physical activity were the strongest contributors. The association between smoking and mortality tended to be further increased in the group with ADs versus the group without such disorders (HR 2.10, 95% CI 1.60 to 2.75 and HR 1.82, 95% CI 1.56 to 2.12, respectively).

**Conclusions** This study suggests potential links between ADs, age and mortality among women with 50 years of follow-up, but does not provide definitive conclusions due to the borderline significance of the results.

## INTRODUCTION

Worldwide, mental disorders are among the top 10 leading causes of morbidity. Major depressive disorder (MDD) and anxiety

**STRENGTHS AND LIMITATIONS OF THIS STUDY**

⇒ Strengths include a follow-up time of five decades, longitudinal design allows for a comprehensive examination of trends and associations over an extended period.

⇒ The inclusion of a representative sample of middle-aged women in Gothenburg with high participation rates enhances the generalisability of the findings.

⇒ The study considers various confounding factors such as smoking, alcohol consumption, physical activity, socioeconomic status and educational level, which previous investigations often neglected.

⇒ A psychiatric investigation based on DSM criteria rather than self-assessed symptoms at baseline ensures the accuracy of the disorders.

⇒ Limitations are the relatively small sample size and an exclusively female population.

disorders (ADs) are the most significant contributors to this, affecting more than half a billion people and are over-represented among women.[1 2] The Swedish National Board of Health and Welfare states that persons with mental disorders have an increased risk of incidence as well as mortality in somatic disorders, compared with people without mental illness.[3] However, the association between mental disorders and mortality is still debatable. In a Danish population-based cohort study, it was seen that mental disorders were associated with a higher mortality rate, regardless of type of mental disorders and to premature death, both for men and women.[4] Also, comorbidity between mental disorder and general medical condition was associated to shorter life expectancy, compared with have one of the conditions only.[5] Denollet *et al* followed 5000 healthy Dutch women for 10 years.[6] They found that anxiety symptoms predicted premature all-cause and cardiovascular mortality.[6] A later study distinguished between anxiety symptoms and ADs

and followed persons for 11–19 years.[7] No prediction of mortality, whether in women or men, was found.[7] A Danish register study with 9 years of follow-up showed that ADs significantly increased mortality risk, both in women and men.[8] Cardiovascular disease and suicide are both potential contributors to increased mortality among people with MDD and AD.[9–11] A variety of factors associated with poor health, such as hypertension, obesity and excess smoking, have also been linked to common mental disorders.[12–16] The challenge remains to determine whether depression/anxiety at baseline is associated with later smoking, or if smoking is associated with later depression/anxiety.[17]

Two meta-analyses from 2015 and 2016 that studied the association between mortality and AD found conflicting results.[18 19] One of the meta-analyses concluded that there was an association between mortality and AD, while the other found no such association. The diversity in results could be due to several reasons. Including psychiatric inpatients tended to increase the association,[18] while studying a community sample and adjusting for comorbid AD/MDD had the opposite effect.[19] Further, the identification of the disorders differed, with the majority based on medical records in one study[18] and self-reported symptoms in the other.[19] However, the follow-up time was a median of 10 years in one of the meta-analyses but less in the other, which limits the evaluation of the long-term effect. MDD, on the other hand, has repeatedly been associated with increased all-cause mortality, especially among men.[20–22] In the Stirling County Study, Murphy *et al* showed an association between depression and premature death among men (HR 2.6, 95% CI 1.4 to 4.9) in 10 years of follow-up. However, this was not seen among depressed women (HR 1.4, 95% CI 0.6 to 3.2).[23]

In 1968, the Prospective Population Study of Women in Gothenburg started, including middle-aged women.[24] Because of its rather extensive content, the repeated follow-up examinations and information regarding mortality during more than 50 years, it was possible to study the associations between disorders and mortality from middle age to the elderly. Given the contradictory results in previous research regarding AD and mortality, the aim was primarily to investigate a possible association between all-cause mortality and AD, secondarily MDD and comorbid AD/MDD, respectively. Another aim was to analyse the risk and health factors which influence such a possible association.

## METHODS AND MATERIAL

The Population Study of Women in Gothenburg includes a representative sample of 1462 women aged 38, 46, 50, 54 and 60 years old who were invited to a medical examination in 1968–1969. The participation rate was 90.1%.[24] A total of 899 participants were systematically sampled from the four youngest age groups according to birth date and invited for a psychiatric examination. Ninety-nine participants declined the invitation, and 22 participants were excluded because of incomplete data due to language difficulties, psychosis or intellectual disability, leading to a total of 778 participants who fulfilled the psychiatric examination and were included in this study (see flow chart in online supplemental appendix figure 1). In this subsample, the participation rate was 89% of the invited subjects in the general population. Of those, 22 women were excluded from the present study because of incomplete data due to language difficulties, psychosis or intellectual disability. A comparison between non-participants and participants showed more single women among the non-participants. Still, no differences were found concerning mental disorders or the need for hospitalisation because of them.[25 26] The psychiatric investigation was 1–2 hours long and was conducted using a comprehensive structured questionnaire.

### ADs and MDD at baseline

A psychiatrist performed the psychiatric examination through a face-to-face interview.[27] The findings from the psychiatric investigation were later revised by scrutinising items in the interview questionnaire relevant to a DSM-III MDD diagnosis.[25] Cases of AD were identified from the dataset using algorithms based on the DSM-IV criteria. AD included social phobia, panic disorder, agoraphobia, generalised anxiety disorder (GAD) and obsessive-compulsive disorder (OCD). Identifying cases of post-traumatic stress disorder from items in the dataset was impossible. Specific phobias were excluded due to their generally limited clinical importance.

### All-cause mortality rate

Information regarding participants' mortality was collected from the Swedish Cause of Death Register. The mortality rate was observed from the first examination date to 31 December 2019.

### Statistics

Baseline characteristics were described using $\chi^2$ and independent t-test. The association between the baseline characteristics in the women with and without AD/MDD was measured by the relative risk (RR) and 95% CI.

Poisson's regression models were used to analyse associations with mortality, and the effect of assumed risk factors on mortality risk is reported as HR and 95% CI. This model used is more accurately designated 'Poison piecewise exponential model' but will be abbreviated as 'Poisson model' in the text.

The follow-up time of each individual is split into many short intervals of equal length. For each interval, an event outcome variable is defined, coded 0 for not dead and 1 if not dead in that time interval. This structures survival data as a time-to-event dataset that will for all practical purposes give identical results whether analysed in a Cox model or a Poisson model. An advantage with Poisson models is that age and calendar time can be modelled as continuous risk factors, making it easy to test for interaction with regard to age or

**Table 1** Baseline characteristics of the different groups of women with AD solely, MDD solely, comorbid AD/MDD and no AD or MDD, at the examination year 1968–1969

| Characteristics, mean (SD) | AD solely n=135 | MDD solely n=32 | Comorbid AD/MDD n=25 | No AD or MDD n=586 |
|---|---|---|---|---|
| Age (years) | 47.6 (4.4) | 47.8 (5.1) | 47.2 (5.3) | 47.9 (4.5) |
| Economic status†, 1000 SEK | 16.2 (8.5) | 16.6 (12.2) | **12.0 (4.4)*** | 16.1 (9.8) |
| Number of cigarettes per day among smokers | 10.5 (6.0) | **14.1 (7.9)*** | 11.2 (8.3) | 10.2 (6.0) |
| BMI, kg/m$^2$ | 24.4 (4.3) | 23.7 (2.6) | 24.2 (3.4) | 24.1 (3.7) |
| Characteristics, Prevalence (%) | | | | |
| Education 6 years or more, n (%) | 32 (23.7) | 6 (18.8) | 3 (12.0) | 153 (26.1) |
| Marital status, married | 102 (75.6) | **19 (59.4)*** | 22 (88) | 483 (82.4) |
| Physical activity Sedentary leisure time‡, n (%) | 27 (20.6) | 4 (12.5) | **8 (32.0)*** | 95 (16.2) |
| Smoking, n (%) | **71 (52.6)**** | 16 (50.0) | 13 (52.0) | 226 (38.8) |
| Economic status (in 10% lowest income category), n (%) | 13 (9.6) | 2 (6.3) | 1 (4.2) | 55 (9.5) |
| Regular alcohol use, n (%) | | | | |
| Beer | **49 (36.2)**** | 5 (15.6) | 5 (20) | 162 (27.6) |
| Wine | 8 (5.9) | 0 (0) | 0 (0) | 33 (5.6) |
| Spirits | 1 (0.7) | 0 (0) | 0 (0) | 8 (1.4) |
| BMI>30 | 13 (9.6) | 1 (3.1) | 1 (4.0) | 44 (7.5) |
| BMI<19 | 8 (5.9) | 1 (3.1) | 2 (8.0) | 28 (4.8) |

The significance is tested for the chosen group compared with the group with neither AD nor MDD.
*p<0.05, **p <0.01.
†Annual income divided by number of family members.
‡<4 hours/week. Values in the cell 'physical activity sedentary leisure time' and 'AD solely n=135' mean that 27 women belong to the sedentary category and they comprise 20.6% of those in the AD solely group (5 are missing data on physical activity). The values show the number of persons with a sedentary lifestyle and their proportion in each AD/MDD category.
AD, anxiety disorder; BMI, body mass index; MDD, major depressive disorder.

calendar time, that is, to test if the effect of a certain (assumed) risk factor varies by age or over calendar years. The HR (95% CI) measures the RR of death at any arbitrarily chosen time interval that, in the absence of interaction, is assumed to be a constant. The risk time was measured from baseline to the time of death or 31 December 2019. The median follow-up time was 36.9 years (minimum 0.33, maximum 51.3 years). The total risk time was 27 260 years and subdivided into quarters of a year (91.3 days).

When analysing the association between mortality and all AD and all MDD, covariates were sequentially added (table 1). Still, each covariate was tested in a separate model with age as the only covariate, as small sample sizes

**Table 2** Associations between all cases of AD, all cases of MDD and mortality rate in different adjustment models

| Covariates | AD n=160 HR (95% CI) | MDD n=57 HR (95% CI) |
|---|---|---|
| Age | **1.25 (1.04 to 1.49)*** | **1.21 (1.11 to 1.12)*** |
| Age, smoking, physical activity and alcohol use (beer) | **1.20 (1.00 to 1.44)*** | 1.18 (0.90 to 1.56) |
| All of the above, economic status, married, education, BMI>30 and BMI<19 | 1.17 (0.98 to 1.41) | 1.19 (0.90 to 1.57) |
| All of the above and depression | 1.16 (0.96 to 1.39) | – |
| All of the above and anxiety | – | 1.15 (0.86 to 1.52) |

Bold figures: statistically significant.
*p<0.001
AD, anxiety disorder; BMI, body mass index; MDD, major depressive disorder.

limited the number of covariates that it would be meaningful to enter simultaneously (table 2).

Statistical significance was accepted at p<0.05 in all analyses except tests of interaction, which was accepted at p<0.10. SPSS V.29 and Stata V.17 were used to perform statistical analysis.

## Potential confounders

To reduce the effect of potential confounders, the following covariates, with known influence on mortality, were used: age (in years), education (over or under 6 years), economic status (defined as annual income divided by number of family members), alcohol consumption (from never to everyday use of wine, beer and liquor, separately, and 'regularly' is defined as several days per week or more), smoking (current or non-current), physical activity ('sedentary leisure time' is defined as <4 hours/week)[28] and body mass index (weight in kilogram divided with the square of height in metres). Lower educational attainment and income levels have been associated with higher mortality rates[29] and marital status can influence mortality, with married individuals experiencing lower mortality rates compared with those who are unmarried.[30]

All covariates except age were dichotomised in the regression models. These data were collected in interviews or measured at baseline.

## RESULTS
### AD, MDD and comorbid AD/MDD

One hundred and sixty participants (20.6%) had one or more AD, and 57 (7.3%) had MDD in the psychiatric examination 1968–1969. Of these, 25 participants (3.2%) had comorbid AD/MDD. The numbers of those with AD solely and MDD solely are presented in table 1.

The distribution of disorders among AD was GAD n=71 (9.1%), agoraphobia n=67 (8.6%), panic disorder n=58 (7.4%), social phobia n=51 (6.6%) and OCD n=28 (3.6%). Of the 135 participants with AD, 85 had one disorder, 48 had two disorders, 17 had three disorders, 7 had four disorders and 3 had all five disorders.

### Baseline characteristics

Baseline characteristics are shown in table 1. Participants with AD solely (ie, excluding participants with comorbid AD/MDD) did, to a greater extent, smoke and consume more alcohol (beer) than those without any disorder. Participants with comorbid AD/MDD had significantly more sedentary leisure time and lower economic status. Participants with MDD were, to a lesser extent, married.

### Smoking, beer consumption and physical activity (measured at baseline)

Having AD without MMD significantly increased the RR of being a smoker (RR 1.38, 95% CI 1.13 to 1.67, p<0.01) and regularly consuming beer (RR 1.35, 95% CI 1.05 to 1.75, p<0.05), compared with not having any of the disorders. Having MDD without AD increased the RR of being a heavy smoker (>13 cigarettes/day) (RR 2.23, 95% CI 1.20 to 4.33, p<0.05). Having comorbid AD/MDD significantly increased the RR of having a sedentary leisure time (RR 1.97, 95% CI 1.08 to 3.59, p=0.05).

### All cases of AD and MDD and all-cause mortality rates

Of the total 778 participants, 720 died before 1 January 2020. A multivariate regression survival model was used to examine the associations between all cases of AD, all cases of MDD and mortality (table 2). In the full model, there was no association between AD and mortality. In a corresponding analysis of all cases of MDD, there was a significant association when adjusted for age.

When restricting the analysis (full model including all AD and all MDD) to deaths occurring before the age of 80, AD became a significant risk factor (HR 1.49, 95% CI 1.13 to 1.98, p<0.01). This association was even stronger in the age range of 65–80 years (HR 1.70, 95% CI 1.26 to 2.29, p<0.001).

To test the significance of interaction for the effect of AD on mortality in different age intervals, one model with three separate AD-effect parameters was calculated for the age groups 38–64, 65–79 and 80–101. This model was compared with a base model assuming constant AD-effect over the ages. The difference in overall model $\chi^2$ between the two Poisson regression models was 8.2 with df=2 and p=0.02.

An age-adjusted model with only two AD-effect parameters, the first for age 65–79 and the second for the younger and older age intervals pooled, yielded $\chi^2$=9.3, df=1 and p=0.002 compared with the base model. A fully adjusted model yielded $\chi^2$=7.94, df=1 and p=0.005.

When analysing the relationship between all-cause mortality and MDD, no significant associations were found. However, it is worth noting that the incidence rate ratio (IRR) exhibited a higher magnitude compared with AD, although with somewhat less precision.

### AD solely, MDD solely, comorbid AD/MDD and all-cause mortality rates

When assessing single risk factors (with age included), a significant association with mortality was found among women with solely AD who smoked. A similar trend was noted in cases of comorbid AD/MDD, as seen in table 3.

In a regression analysis, when adjusted for age, the increased mortality risk associated with smoking did increase further among those having AD, compared with not having AD, HR 2.28 and 1.71, but with overlapping 95% CIs 1.64 to 3.17 and 1.44 to 2.02, respectively, p=0.096. The corresponding difference in mortality risk associated with regular physical activity was negligible, with HR 0.77 (95% CI 0.57 to 1.03) and HR 0.73 (95% CI 0.62 to 0.86), p=0.95.

von Below A, et al. BMJ Open 2023;13:e075471. doi:10.1136/bmjopen-2023-075471

**Table 3** Associations between single covariates and the mortality rate in the total sample and the groups with AD solely, MDD solely and comorbid AD/ADD. All models include age

| Covariate | Total sample n=778 HR (95% CI) | AD solely n=135 HR (95% CI) | MDD solely n=32 HR (95% CI) | Comorbid AD/MDD n=25 HR (95% CI) |
|---|---|---|---|---|
| Age | **1.13 (1.12 to 1.14)***** | **1.13 (1.11 to 1.15)***** | **1.14 (1.09 to 1.19)***** | **1.17 (1.11 to 1.23)***** |
| Education ≥6 years | 0.88 (0.74 to 1.05) | 1.00 (0.62 to 1.60) | 1.20 (0.46 to 3.19) | 0.30 (0.04 to 2.13) |
| Physical activity: Regular exercise | **0.73 (0.63 to 0.85)***** | 0.71 (0.49 to 1.04) | 0.79 (0.38 to 1.66) | 1.02 (0.40 to 2.63) |
| Smoking | **1.82 (1.56 to 2.13)***** | **1.86 (1.24 to 2.79)**** | 2.12 (0.95 to 4.72) | **5.18 (1.57 to 17.14)**** |
| Beer consumption | 0.93 (0.84 to 1.03) | 0.83 (0.66 to 1.05) | 0.70 (0.38 to 1.29) | 1.85 (0.76 to 4.49) |
| BMI<19 | **1.66 (1.20 to 2.31)***** | 2.13 (0.98 to 4.62) | 1.57 (0.17 to 14.07) | 2.38 (0.19 to 29.8) |
| BMI>30 | 1.31 (1.00 to 1.72) | 0.88 (0.47 to 1.67) | 2.07 (0.25 to 16.88) | 0.60 (0.07 to 4.86) |
| Married | 1.00 (0.84 to 1.22) | 0.97 (0.62 to 1.50) | 1.29 (0.54 to 3.07) | 1.08 (0.30 to 3.97) |
| Economic status (lowest decentile) | **1.37 (1.07 to 1.77)*** | 1.43 (0.82 to 2.49) | 0.69 (0.22 to 2.23) | 0.37 (0.05 to 2.86) |

*p<0.05, **p<0.01, ***p<0.001
AD, anxiety disorder; BMI, body mass index; MDD, major depressive disorder.

## DISCUSSION
### Main findings
In this 50-year study of middle-aged women, several significant findings emerged. Women with solely AD were more likely to be smokers and beer consumers compared with those without AD or MDD. Notably, women with AD (all cases) exhibited a significantly increased mortality rate when considering age, smoking, physical activity and beer consumption as covariates in the model. However, this association did not reach statistical significance in the full model. Conversely, women with MDD (all cases) displayed a non-significant trend towards an increased mortality rate in both models. When analysing deaths specifically within the age range of 65–80 years, a strong and significant association was found between AD and increased mortality rate in. Intriguingly, no such associations were identified for the age intervals of 38–64 and 80–101, indicating a significant interaction effect for AD-age-mortality. Notably, no analogous pattern was observed concerning the interaction between MDD-age-mortality.

### Comparison with other studies
The lack of significant overall association between MDD and all-cause mortality is in contrast to previous findings,[20–22] with some exceptions. Holwerda *et al* did find an association, but only among elderly men.[22] Eaton *et al*, on the other hand, did not find any association, and the lack of this is suggested to be primarily due to a long follow-up time (27 years). Symptoms of MDD could be disguised signs of poor health. In the short term, this could lead to an increase in mortality risk that later would subside.[31] However, if this was applicable in our study, it would have been revealed in our sample when analysing time intervals, which was not the case.

Our sample of women with MDD may have been too small, which may explain the lack of significant findings

in this group. It is notable, however, that in most of our analyses of MDD and mortality, the HRs were similar to those in the analyses of AD. For example, in the 80–101 age range, the HR for MMD was 1.39 (95% CI 0.98 to 1.98).

The finding of a reduced association between AD and mortality rate when the full model was applied is consistent with previous research.[19 21 31] The increased mortality rate among women with AD in the age range 65–80 is not readily comparable to earlier studies due to the difference in follow-up time. One interpretation of this finding could be that AD is associated with mortality only in ages >65 years. This contradicts previous results, where AD in older women has not been associated with increased mortality but concurs with our finding of no association after age 80.[22 32] Another interpretation is that AD that lasts for an extended period of time, increases vulnerability and may impede the ability of individuals to manage their health and access necessary medical care, which can further contribute to an increased morbidity and mortality risk.

Smoking is an established risk factor for mortality. The association between smoking and mortality tended to increase further in the presence of AD and comorbid AD/MDD. A different pattern in smoking habits could explain this. We found that smoking was more common among women with AD. Another study reported that persons with AD, to a lesser extent, cease daily smoking. If they do, the duration of smoking is longer.[33] Our study did not investigate the exposure time. Still, if our sample is comparable to that study, and the women had a longer duration of smoking, this could be one possible contributing cause to the late increase in mortality.

A meta-analysis of smoking and mortality found that the mortality rate decreased among female smokers after

75 or 80 years.[34] This is consistent with our findings of reduced mortality after 80 years among women with AD, where 53% were smoking at baseline. One possible explanation is that after the age of 80 years, subjects susceptible to harmful smoking effects may have died already. Another possible reason might be that after the age of 80 years, other causes of death increase and compete with those related to smoking and AD.

### Strengths and weaknesses

This study exhibits several strengths, including its extended five-decade longitudinal design, a representative sample of middle-aged women in Gothenburg with high participation rates, and a comprehensive consideration of various confounding factors. Importantly, prior investigations reporting increased mortality rates in ADs did not control for covariates such as smoking, alcohol consumption, physical activity, socioeconomic status or educational level, which could have influenced their findings.[4 5 8]

Also, many studies investigating AD and MDD have used various self-administered questionnaires, measuring symptomatology rather than the frequency of a disorder. This could potentially overdiagnose or underdiagnose the psychopathology, and the reported symptoms could originate from a somatic condition. In this study, however, the disorder was assessed by a psychiatrist, making the results more accurate. Furthermore, studies tend to investigate associations either among persons with AD or MDD, which could be misleading due to the high comorbidity. In contrast, in the current study, the participants with comorbid AD/MDD were examined as a separate group.

The study has some limitations. First, the research sample was relatively small, especially regarding the group having MDD and comorbid AD/MDD. While there is a potential for complex causal mechanisms linking depression status, lifestyle factors such as smoking and alcohol consumption, and mortality risks, our study is primarily focused on examining associations and variations. Introducing complex models would not enhance our analysis, as our dataset lacks the necessary scope to support such an approach. Another limitation was that the study sample only included women. Furthermore, somatic diseases among the participants were not taken into account. It is possible that patients with AD could have a higher rate of somatic disease, which in turn could affect the mortality rate. Lastly, it is important to acknowledge that the decision made at the outset to confine the evaluation of baseline risk factors and their outcomes, as well as to assess the point prevalence of AD and MDD (not the lifetime prevalence), represents a limitation in this study. However, it is worth noting that this limitation may not necessarily introduce biases, especially when taking into account the specific research focus of our study. However, the onset of AD is, above all, in early adolescence to young adulthood and often has a long duration, which is why we can consider that the participants who meet the criterion for AD in middle age have had the disorder for a long time.

Correspondingly, most of those with the preponderance to incur the condition should have received it by the time of the baseline examination. The same is true for MDD, which has a median onset in young adulthood.[35]

### The significance of the study, unanswered questions and future research

This study adds to the previous findings showing that AD is not convincingly associated with mortality over the entire age interval studied. The same applies to comorbid AD/MDD. The MDD group was small, however, and the results concerning MDD and mortality are, therefore, inconclusive. The increase in mortality among those with AD aged 65–80 years is a novel finding that needs to be confirmed in future studies. Considering the high prevalence of comorbid AD/MDD cases, future studies on the association between mortality and AD or MDD should include both disorders. Future research should also focus on effective methods to promote a healthier lifestyle among persons with mental disorders especially regarding cessation of smoking and reduction of heavy alcohol consumption to counteract the increased likelihood of developing comorbid medical conditions such as cardiovascular disease.

**Contributors**  The study was conducted as part of AvB's intern research project. DH, TH and CB supervised the work. TH was responsible for the AD and MDD diagnostic procedures. AvB analysed the data and prepared the draft manuscript with the support of all the other authors, DH, TH and CB. VS devised the statistical analysis and contributed to the interpretation of it. All the authors approved the final manuscript and revision. DH act as guarantor.

**Funding**  This work was supported by Grants from the Swedish Research Council for Health, Working Life and Welfare—FORTE 2007-1958 and grants from the Swedish State under the agreement between the Swedish Government and the county council, the ALF agreement GBG-68771.

**Competing interests**  None declared.

**Patient and public involvement**  Patients and/or the public were not involved in the design, or conduct, or reporting, or dissemination plans of this research.

**Patient consent for publication**  Not applicable.

**Ethics approval**  This study involves human participants and the Regional Ethical Committee in Gothenburg approved the study (258-16 T853-16). Participants gave informed consent to participate in the study before taking part.

**Provenance and peer review**  Not commissioned; externally peer reviewed.

**Data availability statement**  Data are available on reasonable request. Data are available on request.

**ORCID iDs**
Cecilia Björkelund http://orcid.org/0000-0003-4083-7342

Dominique Hange http://orcid.org/0000-0003-1114-4440

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
