## [Reviewer comments · BMJ Open]

ARTICLE DETAILS

TITLE (PROVISIONAL)	Association between Anxiety and Depression and all-cause Mortality: a 50-year follow-up of the Population Study of Women in Gothenburg, Sweden
AUTHORS	von Below, Amanda; Hällström, Tore; Sundh, Valter; Björkelund, Cecilia; Hange, Dominique

VERSION 1 – REVIEW

REVIEWER	Yu , Yongfu Fudan University, Department of Biostatistics
REVIEW RETURNED	02-Jun-2023

GENERAL COMMENTS	It is my pleasure to review this important study. Several papers from John Joseph McGrath, Oleguer Plana-Ripoll, and his colleagues have examined the association between mental disorders and premature mortality in the Danish population (Lancet 2019, PLoS Medicine 2022, JAMA Psychiatry. 2022, Psychiatry Research 2023). The author should clearly state what this study added. The authors may need to consult a biostatistician about data analysis and result presentation. I have a few suggestions that may improve the manuscript: 1. Risk ratio rather than odds ratio should be used2. Why did the authors use both Poisson regression and Cox regression?3. Table 1, the authors only list BMI>30 and BMI<19, how about 19-30?4. Confounder definition: “alcohol consumption (five level item, from never to everyday use of wine, beer and liquor, separately), physical activity (four level item, from a sedentary leisure time to regular physical exercise)”. However, the data presented in Table 1 is not by five levels of alcohol consumption and four levels of physical activity.5. When I read table 1, I suspect there is a sparse data problem.6. Result section, it is not common to present Chi-square and df7. What is the rationale for confounder selection?
--

REVIEWER	Seet, Vanessa Institute of Mental Health Centre for Evidence-Based Practices in Mental Health Care
REVIEW RETURNED	08-Jun-2023

GENERAL COMMENTS	Thank you for the opportunity to review this manuscript. The topic is an important one, and the results of the study have significant implications on the study of physical and mental illness co-morbidity. However, there are certain issues regarding the reporting of the statistical methodology and results that limit the
---

	readability of the manuscript. I hope my comments help clarify and improve this work.  - Page 3 (of 19): one of the objectives was to examine specific risk and health factors that may mediate the association between AD and/or MDD and all-cause mortality. However, the only mention of something along the lines of a mediation analysis was in Page 10, and it was reported as tests of interaction, not a sequential mediation analysis - Page 4, line 30: please change to “over / for 11-19 years”. Currently, it implies the persons are aged 11-19 years, which I do not think is the case. - Page 4, lines 47-49: please include the citations of the two meta-analyses - Page 6, lines 38-41: (and also with reference to Tables 2 and 3) was the reference level women without either AD or MDD (ie without any disorder)? Or were the separate models tested with different reference levels – for example, in table 3, are the HRs for those with solely AD with reference to those without either AD or MDD, or those without AD only (but who may have an MDD diagnosis)? Please clarify. - Page 7: what was the rationale behind adding these covariates in the models, especially the sociodemographic ones? Please include a few statements and citation on the associations these covariates may have with mortality to justify their inclusion into the model, especially since there is a limit in how many can be included due to the small sample size - Table 1: there seems to be a discrepancy in the table figures – for instance, please see the cell under AD solely for “Physical activity Sedentary leisure time) – the % derived seems different from that of the other groups. Moreover, to clarify, is this row depicting the prevalence of participants engaging in excessive sedentary leisure time?
--	--

REVIEWER	Okui, Tasuku Kyushu University
REVIEW RETURNED	20-Jun-2023

GENERAL COMMENTS	The manuscript seems to be valuable by using a cohort study with a long follow-up time, but some points need to be clarified.  1. Abstract; Could you write the number or prevalence of AD in this study? In addition, how about showing the results of major depressive disorder if it is another outcome? Moreover, age and AD status mean those at the time of 1968-69? Furthermore, the meaning of “When examining age during risk time as separate intervals” is uncertain as an English, while readers can understand those are descriptions about Poisson regression after reading the main body of the manuscript. 2. Methods and Materials; It is written that the study included women aged 38, 46, 50, 54, and 60 years old. In this case, why the age range 65-80 years which is written in Abstract existed in the data is uncertain. Could you add an explanation of the age group? 3. Statistics, line 2; “95% confidence interval” is an accurate expression for the words. 4. Statistics; Is it right that logistic regression was used for calculating OR? In addition, did you calculate HR from Poisson regression by separating study periods by each of event? More clarification is needed for the methods.
---

	5. Results; Although it is written that 1468 subjects were included in the setting in the Abstract, it is written that 778 participants were included in the Results. Are they consistent? In addition, how about showing a flowchart of selecting participants? 6. Discussion; It is considered that AD patients tend to have other chronic diseases and it could have affects the mortality rates of those patients. I think that this point needs to be discussed as a limitation.
--	---

REVIEWER	Gasparini, Antonio London School of Hygiene and Tropical Medicine
REVIEW RETURNED	17-Jul-2023

GENERAL COMMENTS	The manuscript is well written and structured, with a clear research hypothesis and a compact set of results. My main concerns relate to their interpretation, though, which is somewhat stretched and fails to appropriately acknowledge the weak evidence and mostly borderline significant estimates. Specific comments are added below.  1. Abstract, sections on Setting and Participants: the authors mention that the study concerns a sample of 1,468 women, but apparently this is the full cohort, out of which only 899 were invited to participate and eventually only 778 were included in the study. This is misleading and must be corrected. 2. Abstract: in the section Primary and secondary outcome measures, the authors should reverse the order of the stated associations, indicating that they assessed the relationship between the conditions and all-cause mortality, not the other way round. 3. Abstract and corresponding text: it is unclear why the authors report first partially-adjusted estimate and then the fully-adjusted one. I suspect it is because the former still shows a significant association at the 95% level of confidence. While I am not sure what the Journal's policy is, there is nothing special about the 95% choice, and the two estimates indicate pretty much the same evidence. I suggest reporting only the latter or otherwise justify the choice. 4. Abstract and corresponding text: he authors claim that "no associations were found in the younger or older age interval". However, similarly, these estimates are somewhat lower and do not reach significance at the 95% level confidence, but there can still be an indication of an effect. I suggest rephrasing and stating that. 5. Abstract and corresponding text: in the final part, the authors report estimates from stratified analyses, however, without reporting the results from significance tests of such differences. 6. Abstract and corresponding text: the conclusions are too strong, considering the borderline significant results. Please rephrase. 7. Page 6, lines 32 and below: there is some confusion about the effect summaries used in the study. The Poisson model reports estimates of incidence relative risk (IRR, or rate ratio, depending on if an offset is included), not a hazard ratio. The latter is reported by Cox proportional hazard models. I understand it is only a matter of terminology, but it is nevertheless important. 8. Page 6, lines 32 and below: as mentioned above, it is unclear if the (log) person-years in each stratified observation used in the Poisson regression are included as an offset, as they should. Please clarify. 9. Page 7, lines 3:18: calendar year does not seem to be included as a potential risk factor. While it is unclear if it can have any effect
--

	at all, I suggest including it, and this would be relatively simple in the Poisson model proposed by the authors. 10. Page 7, lines 3:18: it is unclear if the measures of potential confounders were collected only at baseline or also during the follow-up. Please clarify. 11. Page 7, line 35: the authors mention that 778 women were included in the study, but above they state that the total sample was 1,468. Please clarify and if needed discuss potential selection problems. 12. Page 8, line 43: please clarify in this section that this analysis refer to associations between variables measured at baseline only. 13. Page 8, line 55 and the following text: similarly to what discussed in the first comments above, the description is misleading, as it puts too much emphasis on partially-adjusted models instead that stating that the evidence is much weaker when controlling for all the relevant confounders. Please rephrase. 14. Page 9, lines 37 and below: the description of the model including an interaction should be moved to Material and Methods, explaining also its purpose. 15. Page 9, line 53: the authors state here that “No associations were seen when analysing MDD and all-cause mortality”, when in fact the IRR is actually higher than for the association with AD, only less precise. Please correct the statement. 16. Page 10, Table 3: it is unclear if the models reported here are controlled for age plus a single factor at each time. If so, this is misleading, and the authors should primarily report and discuss the estimates for single conditions with fully-adjusted models. 17. Page 10, lines 40-49: the interpretation of the results of models with interactions is misleading, as based on cherry-picking and commenting on estimates affected by weak evidence of an association. The usual approach in interpreting the results of multiple tests on interaction requires being stricter in ruling out associations arising by chance, and in this case there seems to be limited evidence in the data. Please change the text accordingly. 18. Page 11, lines 5-29: this section should contain a summary of the results, not a simple list of them. Please change the text accordingly, highlighting the salient results. 19. Page 11, line 36: as commented earlier, the analysis does not suggest an absence of association between MMD and mortality, given that the point estimate is actually higher than for AD. This study just seems not powered enough to identify precise results on this association. This issue is actually commented on in the second part of the section, which should be the main interpretation reported here. 20. Discussion: a critical limitation of this study, not clearly acknowledged anywhere in this section, is that both AD/MMD diagnoses and the other risk factors (e.g., smoking, physical activity, etc) were only measured at baseline, sometimes decades before the outcome. This can possibly lead to biases and wrong conclusions, and should be made clear. 21. Discussion: similarly, there is no mention to the potential complex causal mechanisms between depression status, lifestyle factors such as smoking and alcohol consumption, and mortality risks. While this study only aims to look at association, a more comprehensive discussion on the causal aspects is required.
--	--

VERSION 1 – AUTHOR RESPONSE

Reviewer: 1

Several papers from John Joseph McGrath, Oleguer Plana-Ripoll, and his colleagues have examined the association between mental disorders and premature mortality in the Danish population... The author should clearly state what this study added.

Author response: This is a valid point. As suggested by the reviewer, we have stated what two of these studies added in the first paragraph of the introduction.

The authors may need to consult a biostatistician about data analysis and result presentation. Author response: One of the authors, VS is a biostatistician with great experience in data analysis and result presentation.

1. Risk ratio rather than odds ratio should be used

Author response: OK, makes sense for descriptive purposes. We have changed into a risk ratio. See section heading, "Smoking, beer consumption and physical activity (measured at baseline)".

2. Why did the authors use both Poisson regression and Cox regression?

Author response: That is a misunderstanding. We only use Poisson regression in this work and mention Cox regression, which is the model most often used for this type of time-to-event data, to explain that both models return the same results. Also, see a separate document about the use of Poisson regression models.

3. Table 1, the authors only list BMI>30 and BMI<19, how about 19-30?

Author response: The percentage can be calculated from the low and high BMI groups.

BMI = 19 – 30, 100-(9.9+6.1)% , 100-(3.1+3.1)% , 100-(4.0+8.0)% , 100-(7.5+4.8)%

We have not changed in Table 1.

4. Confounder definition: "alcohol consumption (five level item, from never to everyday use of wine, beer and liquor, separately), physical activity (four level item, from a sedentary leisure time to regular physical exercise)". However, the data presented in Table 1 is not by five levels of alcohol consumption and four levels of physical activity.

Author response: This is related to point 5. Low sample size makes the pooling of small categories analysis more robust. We have rewritten it under the section heading "Potential confounders" and clarified it in Table 1.

5. When I read table 1, I suspect there is a sparse data problem.

Author response: Agree, not a large total sample and a low proportion of individuals with a risk factor make any conclusions less reliable. However, it may still contribute to the general knowledge of the relation between the studied factors. We have corrected the calculations in Table 1.

6. Result section, it is not common to present Chi-square and df

Author response: Yes, but not always necessary. If the aim had been to find the best prediction models for the combined effect of all effect variables it would be necessary to report the overall goodness-of-fit between the observed outcome and predicted outcome from the model we had constructed, for example, model logLR ChiSq and df or AUC or AIC etc. But we don't see this as necessary here as the focus is on the importance of each separate effect variable.

7. What is the rationale for confounder selection?

Author response: We have added: To reduce the effect of potential confounders, the following covariates, with known influence on mortality were used, smoking, alcohol consumption, physical activity, socioeconomic status and educational level. We have added a part of this in the discussion and

Lower educational attainment and income levels have been associated with higher mortality rates (Backlund, Sorlie, & Johnson, 1996) and marital status can influence mortality, with married individuals experiencing lower mortality rates compared to those who are unmarried (Manzoli et al., 2007).

Reviewer: 2

However, there are certain issues regarding the reporting of the statistical methodology and results that limit the readability of the manuscript...

1. Page 3 (of 19): one of the objectives was to examine specific risk and health factors that may mediate the association between AD and/or MDD and all-cause mortality. However, the only mention of something along the lines of a mediation analysis was in Page 10, and it was reported as tests of interaction, not a sequential mediation analysis.

Author response: We don't think that a full mediation analysis is meaningful for data of this size for reason of power. So the question is if it is allowed to use the word mediation in a colloquial sense, like 'have an effect on', in the text and not in a scientific sense. We see no problem with this, it is common to do that also in scientific texts, for example when we talk about linear correlation from a prediction model and mean that r is substantially larger than zero, although in mathematics linear correlation means that $r=1.0$. However, we can substitute 'mediate' for something else that is appropriate, like 'influence' if it is demanded.

2. Page 4, line 30: please change to "over / for 11-19 years". Currently, it implies the persons are aged 11-19 years, which I do not think is the case.

Author response: Thank you for pointing this out, we have changed it to "for 11-19 years".

3. Page 4, lines 47-49: please include the citations of the two meta-analyses

Author response: Thank you for pointing this out. We have changed it by the reviewer's comment.

4. Page 6, lines 38-41: (and also with reference to Tables 2 and 3) was the reference level women without either AD or MDD (ie without any disorder)? Or were the separate models tested with different reference levels – for example, in table 3, are the HRs for those with solely AD with reference to those without either AD or MDD, or those without AD only (but who may have an MDD diagnosis)? Please clarify.

Author response: In table 2 the comparisons are between those with and without AD, and between those with or without MDD. In Table 3, each column defines the sample that is analyzed, so there are no comparisons within the models between AD/MDD categories.

5. Page 7: what was the rationale behind adding these covariates in the models, especially the sociodemographic ones? Please include a few statements and citation on the associations these covariates may have with mortality to justify their inclusion into the model, especially since there is a limit in how many can be included due to the small sample size

Author response: We have added: To reduce the effect of potential confounders, the following covariates, with known influence on mortality were used, smoking, alcohol consumption, physical activity, socioeconomic status and educational level. We have added a part of this to the discussion and

Lower educational attainment and income levels have been associated with higher mortality rates (Backlund, Sorlie, & Johnson, 1996) and marital status can influence mortality, with married individuals experiencing lower mortality rates compared to those who are unmarried (Manzoli et al., 2007).

6. Table 1: there seems to be a discrepancy in the table figures – for instance, please see the cell under AD solely for “Physical activity Sedentary leisure time) – the % derived seems different from that of the other groups. Moreover, to clarify, is this row depicting the prevalence of participants engaging in excessive sedentary leisure time?

Author response: Thank you for pointing this out. We have reviewed the % derived and corrected them. The row describes participants who have excessive sedentary leisure time. We have tried to clarify in Table 1 by adding an explanation.

Cross table between AD/MDD categories and sedentary level:
(5 are missing data on physical activity)

```

+-----+-----+-----+
! Sedentary ! Active ! TOTAL!
+-----+-----+-----+
ADnoMDD ! 27 104 131!
Row % ! 20.6 79.4 100.0!
+-----+-----+-----+
MDDnoAD ! 4 28 32!
Row % ! 12.5 87.5 100.0!
+-----+-----+-----+
ADandMDD ! 8 17 25!
Row % ! 32.0 68.0 100.0!
+-----+-----+-----+
NoADMDD ! 95 490 585!
Row % ! 16.2 83.8 100.0!
+-----+-----+-----+
Total ! 134 639 773!
! 17.3 82.7 100.0!
-----+-----+-----+

```

The values in Table 1 show the number of persons with a sedentary lifestyle and their proportion in each AD/MDD category.

Reviewer: 3

1. Abstract; Could you write the number or prevalence of AD in this study? In addition, how about showing the results of major depressive disorder if it is another outcome? Moreover, age and AD status mean those at the time of 1968-69? Furthermore, the meaning of “When examining age during risk time as separate intervals” is uncertain as an English, while readers can understand those are descriptions about Poisson regression after reading the main body of the manuscript.

Author response: Prevalence numbers of AD, MDD and AD/MDD is added in the abstract.

We have also added “in the psychiatric exam 1968-69” to the sentence: Of those, one hundred sixty participants (20.6%) had one or more AD, and 57 (7.3%) had MDD in the psychiatric exam 1968-69.

2. Methods and Materials; It is written that the study included women aged 38, 46, 50, 54, and 60 years old. In this case, why the age range 65-80 years which is written in Abstract existed in the data is uncertain. Could you add an explanation of the age group?

Author response: Thank you for pointing this out. The age range 65-80 years indicates the age of death. We have clarified the explanation in the abstract to “a significant association between mortality and AD was seen in the group of participants who died at the age of 65-80 years”

3. Statistics, line 2; “95% confidence interval” is an accurate expression for the words.

Author response: The reviewer is correct, and we have changed it by the reviewer's comment.

4. Statistics; Is it right that logistic regression was used for calculating OR? In addition, did you calculate HR from Poisson regression by separating study periods by each of event? More clarification is needed for the methods.

Author response: We only use Poisson regression in this work and mention Cox regression, which is the model most often used for this type of time-to-event data, to explain that both models return the same results. In the Statistics section, the Poisson model is described in more detail. Also, see a separate document about the use of Poisson regression models.

5. Results; Although it is written that 1468 subjects were included in the setting in the Abstract, it is written that 778 participants were included in the Results. Are they consistent? In addition, how about showing a flowchart of selecting participants?

Author response: Thank you for pointing this out. We have clarified the number of participants and added a flowchart of selecting participants.

6. Discussion; It is considered that AD patients tend to have other chronic diseases and it could have affects the mortality rates of those patients. I think that this point needs to be discussed as a limitation.

Author response: That is a valid point. We have added a sentence where we discuss this limitation.

Reviewer: 4

1. Abstract, sections on Setting and Participants: the authors mention that the study concerns a sample of 1,468 women, but apparently this is the full cohort, out of which only 899 were invited to participate and eventually only 778 were included in the study. This is misleading and must be corrected.

Author response: Thank you for pointing this out. The reviewer is correct, and we have clarified the number of participants. The revised text reads as follows in the section Participants "In 1968-69, 899 (out of in total 1462) women from PSWG were systematically selected."

2. Abstract: in the section Primary and secondary outcome measures, the authors should reverse the order of the stated associations, indicating that they assessed the relationship between the conditions and all-cause mortality, not the other way round.

Author response: As suggested by the reviewer, we have reversed the order of the stated associations.

3. Abstract and corresponding text: it is unclear why the authors report first partially-adjusted estimate and then the fully-adjusted one. I suspect it is because the former still shows a significant association at the 95% level of confidence. While I am not sure what the Journal's policy is, there is nothing special about the 95% choice, and the two estimates indicate pretty much the same evidence. I suggest reporting only the latter or otherwise justify the choice.

Author response: We report only the latter.

4. Abstract and corresponding text: the authors claim that "no associations were found in the younger or older age interval". However, similarly, these estimates are somewhat lower and do not reach significance at the 95% level confidence, but there can still be an indication of an effect. I suggest rephrasing and stating that.

Author response: We have rephrased and stating that.

5. Abstract and corresponding text: in the final part, the authors report estimates from stratified analyses, however, without reporting the results from significance tests of such differences.

Author response: We have clarified it under the section heading “Statistics”.

6. Abstract and corresponding text: the conclusions are too strong, considering the borderline significant results. Please rephrase.

Author response: We have rephrased.

7. Page 6, lines 32 and below: there is some confusion about the effect summaries used in the study. The Poisson model reports estimates of incidence relative risk (IRR, or rate ratio, depending on if an offset is included), not a hazard ratio. The latter is reported by Cox proportional hazard models. I understand it is only a matter of terminology, but it is nevertheless important.

Author response: We only use Poisson regression in this work and mention Cox regression, which is the model most often used for this type of time-to-event data, to explain that both models return the same results. As said above, the data we analyze is structured as time-to-event data, and the parameters obtained from Cox regression and Poisson regression are, for all practical purposes, identical. Therefore, it is correct to label the reported effect values as Hazard Ratio. A basic illustration of the similarity of Cox Regression and Poisson Regression can be found at the link <https://pauldickman.com/software/stata/compare-cox-poisson/> Also, see a separate document about the use of Poisson regression models.

8. Page 6, lines 32 and below: as mentioned above, it is unclear if the (log) person-years in each stratified observation used in the Poisson regression are included as an offset, as they should. Please clarify.

Author response: We have clarified the section heading “Statistics”.

9. Page 7, lines 3:18: calendar year does not seem to be included as a potential risk factor. While it is unclear if it can have any effect at all, I suggest including it, and this would be relatively simple in the Poisson model proposed by the authors.

Author response: We don't agree and have not changed it.

10. Page 7, lines 3:18: it is unclear if the measures of potential confounders were collected only at baseline or also during the follow-up. Please clarify.

Author response: Thank you for pointing this out. We have clarified that the time for measures of potential confounders was collected only at baseline.

11. Page 7, line 35: the authors mention that 778 women were included in the study, but above they state that the total sample was 1,468. Please clarify and if needed discuss potential selection problems.

Author response: This is a valid point. We have clarified the number of participants in the first paragraph of the section Results, and also added a flowchart of the participants (see figure S1 in appendix). A comparison of participants and non-participants is mentioned in the first paragraph of Methods and Materials.

12. Page 8, line 43: please clarify in this section that this analysis refer to associations between variables measured at baseline only.

Author response: This has been clarified by adding “measured at baseline” to the heading.

13. Page 8, line 55 and the following text: similarly to what discussed in the first comments above, the description is misleading, as it puts too much emphasis on partially-adjusted models instead that stating that the evidence is much weaker when controlling for all the relevant confounders. Please rephrase.

Author response: We have rephrased.

14. Page 9, lines 37 and below: the description of the model including an interaction should be moved to Material and Methods, explaining also its purpose.

Author response: We are sorry for the confusion. We have moved the heading because we are describing two different sorts of interactions. The interaction described on page 8 concerns age and the interaction mentioned on page 9 is regarding different effects in different risk groups.

15. Page 9, line 53: the authors state here that “No associations were seen when analysing MDD and all-cause mortality”, when in fact the IRR is actually higher than for the association with AD, only less precise. Please correct the statement.

Author response: We have corrected the statement.

16. Page 10, Table 3: it is unclear if the models reported here are controlled for age plus a single factor at each time. If so, this is misleading, and the authors should primarily report and discuss the estimates for single conditions with fully-adjusted models.

Author response: We have tried to explain the models in Table 3.

17. Page 10, lines 40-49: the interpretation of the results of models with interactions is misleading, as based on cherry-picking and commenting on estimates affected by weak evidence of an association. The usual approach in interpreting the results of multiple tests on interaction requires being stricter in ruling out associations arising by chance, and in this case there seems to be limited evidence in the data. Please change the text accordingly.

Author response: We have tried to change the text.

18. Page 11, lines 5-29: this section should contain a summary of the results, not a simple list of them. Please change the text accordingly, highlighting the salient results.

Author response: We have changed the text.

19. Page 11, line 36: as commented earlier, the analysis does not suggest an absence of association between MMD and mortality, given that the point estimate is actually higher than for AD. This study just seems not powered enough to identify precise results on this association. This issue is actually commented on in the second part of the section, which should be the main interpretation reported here.

Author response: We have already written that “Our sample of women with MDD may have been too small” which could be an explanation for the absence of association.

20. Discussion: a critical limitation of this study, not clearly acknowledged anywhere in this section, is that both AD/MMD diagnoses and the other risk factors (e.g., smoking, physical activity, etc) were only measured at baseline, sometimes decades before the outcome. This can possibly lead to biases and wrong conclusions, and should be made clear.

Author response: The decision made at the outset to restrict the evaluation of baseline risk factors and their outcomes, while preventing any misinterpretation as risk factors evolving, is recognized as a moderate limitation. In light of this limitation, it prompts consideration of the possibility of conducting an alternative study that employs more comprehensive data collection methods. It is crucial to explicitly address the temporal gap in measuring both AD/MDD diagnoses and other risk factors, which carries the potential to introduce biases and result in erroneous conclusions. However, it should be emphasized that these biases may not necessarily lead to the misinterpretation of these factors as evolving risk factors over time, especially given the specific research focus of our study. We have added a part to the discussion.

21. Discussion: similarly, there is no mention to the potential complex causal mechanisms between depression status, lifestyle factors such as smoking and alcohol consumption, and mortality risks. While this study only aims to look at association, a more comprehensive discussion on the causal aspects is required.

Author response: Causal analysis would require a much larger dataset due to the limitations of our current sample size. Therefore, we have limited ourselves to examining associations and variations. Attempting to fit complex models would not enhance our analysis as the dataset is not substantial enough to support such an approach, as we have added in the discussion.

VERSION 2 – REVIEW

REVIEWER	Yu , Yongfu Fudan University, Department of Biostatistics
REVIEW RETURNED	04-Oct-2023

GENERAL COMMENTS	Thanks so much for the response. I still have a few questions. 1. If Poisson regression is used, please report incidence rate ratio rather than hazard ratio. I know that Cox regression and Poisson regression would give the same estimate if data were split by events. However, they are two methods and have different assumptions. The other reviewers also have similar concerns. 2. Previously, I suggested that "Risk ratio rather than odds ratio should be used" Author response: "OK, makes sense for descriptive purposes. We have changed into a risk ratio." I am not sure what is descriptive purposes. The authors mentioned they have changed into a risk ratio. However, OR and risk ratio are not the same measure. How did the authors do this?
---

REVIEWER	Okui, Tasuku Kyushu University
REVIEW RETURNED	04-Oct-2023

GENERAL COMMENTS	Thank you for the revision. I confirmed it.
---

VERSION 2 – AUTHOR RESPONSE

Reviewer: 1

1. If Poisson regression is used, please report incidence rate ratio rather than hazard ratio. I know that Cox regression and Poisson regression would give the same estimate if data were split by events. However, they are two methods and have different assumptions. The other reviewers also have similar concerns.

Author response: Here is a comprehensive explanation from our statistician (VS) who has been involved in this study since the early 1990s. They possess an intimate familiarity with the dataset in its entirety and a deep understanding of all the statistical methods necessary to conduct the analyses required for our study. "A very simple explanation of the difference between relative risk and hazard ratio: Hazard ratios differ from relative risks and odds ratios in that RRs and ORs are cumulative over an entire study, using a defined endpoint, while HRs represent instantaneous risk over the study time period. It's possible to approximate a Cox model with a series of Poisson models. That's called a

"piecewise exponential model." You break up the time axis into a series of short time periods. You assume a constant baseline hazard within each individual time period and fit a Poisson model with regression coefficients for covariates assumed to be the same for all periods. A third possibility in this context is the Weibull regression model. The 'Cox', 'Poisson' and 'Weibull' models all produce estimates of what can be described as 'an instantaneous incidence rate ratio', but normally we call it Hazard Ratio for them all. This is not changed by the fact that the models in parts use different underlying assumptions".

To avoid further confusion, we will explicitly state that, when the Poisson model is mentioned in the text, it is an abbreviation for 'Poisson piecewise exponential model'.

2. Previously, I suggested that "Risk ratio rather than odds ratio should be used" Author response: "OK, makes sense for descriptive purposes. We have changed into a risk

ratio." I am not sure what is descriptive purposes. The authors mentioned they have changed into a risk ratio. However, OR and risk ratio are not the same measure. How did the authors do this?

Author response: We use the term 'descriptive purposes' because it is employed within the section on Baseline characteristics. I apologize for the earlier mistake in my point-by-point response to the reviewer, where I incorrectly used the term 'a Risk ratio' instead of the correct term, 'Relative Risk, RR.' This error has already been corrected in the article.